# Relevant Aspects of Titanium Topography for Osteoblastic Adhesion and Inhibition of Bacterial Colonization

**DOI:** 10.3390/ma16093553

**Published:** 2023-05-05

**Authors:** Raquel Rodriguez-González, Loreto Monsalve-Guil, Alvaro Jimenez-Guerra, Eugenio Velasco-Ortega, Jesus Moreno-Muñoz, Enrique Nuñez-Marquez, Roman A. Pérez, Javier Gil, Ivan Ortiz-Garcia

**Affiliations:** 1Bioengineering Institute of Technology, Faculty of Dentistry, Universitat Internacional de Catalunya, Sant Cugat del Vallé, 08198 Barcelona, Spain; 2Faculty of Dentistry, University of Seville, 41009 Seville, Spain

**Keywords:** titanium, roughness, topography, osteoblasts, bacteria, periimplantitis

## Abstract

The influence of the surface topography of dental implants has been studied to optimize titanium surfaces in order to improve osseointegration. Different techniques can be used to obtain rough titanium, however, their effect on wettability, surface energy, as well as bacterial and cell adhesion and differentiation has not been studied deeply. Two-hundred disks made of grade 4 titanium were subjected to different treatments: machined titanium (MACH), acid-attacked titanium (AE), titanium sprayed with abrasive alumina particles under pressure (GBLAST), and titanium that has been treated with GBLAST and then subjected to AE (GBLAST + AE). The roughness of the different treatments was determined by confocal microscopy, and the wettability was determined by the sessile drop technique; then, the surface energy of each treatment was calculated. Osteoblast-like cells (SaOs-2) were cultured, and alkaline phosphatase was determined using a colorimetric test. Likewise, bacterial strains *S. gordonii*, *S. oralis*, *A. viscosus*, and *E. faecalis* were cultured, and proliferation on the different surfaces was determined. It could be observed that the roughness of the GBLAST and GBLAS + AE was higher, at 1.99 and 2.13 μm of Ra, with respect to the AE and MACH samples, which were 0.35 and 0.20 μm, respectively. The abrasive treated surfaces showed lower hydrophilicity but lower surface energy. Significant differences could be seen at 21 days between SaOS-2 osteoblastic cell adhesion for the blasted ones and higher osteocalcin levels. However, no significant differences in terms of bacterial proliferation were observed between the four surfaces studied, demonstrating the insensitivity of bacteria to topography. These results may help in the search for the best topographies for osteoblast behavior and for the inhibition of bacterial colonization.

## 1. Introduction

After Prof. Branemark described the phenomenon of osseointegration, the first dental implants were electropolished with a roughness (Ra) of around 0.15 μm [1,2]. As knowledge of bone fixation progressed, dental implants were subjected to different surface treatments to roughen them by means of acid etching treatments, such as grit blasting with different types of abrasives, by laser, or other methods. This allowed dental implants with a higher roughness to be obtained, which potentially increases the surface area in contact with the bone, making it around eight times greater [3,4,5]. The grit blasting treatments produce a macroroughness, which has been combined with acid etching to achieve a microstructure in the roughness of the titanium. All these treatments were carried out with the aim of promoting the good adhesion, proliferation, and differentiation of osteoblastic cells to achieve good osseointegration [6,7,8,9].

The properties of titanium implant surfaces are an important factor in order to achieve osseointegration as they are involved in the early process of healing after their insertion in alveolar bone. In fact, previous in vitro studies have shown an increment in osteoblast-like cell attachment and activity correlated with an increase in surface roughness [10,11].

Currently, the methods used to increase the roughness of the implants include hydroxyapatite coating, acid-etching, shot blasting, plasma spray, and combinations of these methods [12,13]. In all cases, it has been demonstrated that rough surfaces have an impact on the success of the bone regeneration process, allowing a firmer fixation than those implants with a smooth surface [14,15]. This is related to higher contact between the bone and the implant when comparing to a machined surface, which in turn triggers a stronger bone response [16,17,18]. Moreover, machined surfaces were comparted to different rough surfaces in an animal in vivo study to analyze their effect in the early bone response [18]. The results showed that all the rough implants had an increased bone response when compared to the control. Therefore, the study concluded that modified surfaces promoted faster osseointegration and bone healing [18].

A few years ago, however, it became clear that the adhesion of bacteria to the titanium surfaces was also important due to peri-implantitis [19,20,21,22]. Dental implant manufacturers reduced the roughness and, in some cases, even switched back to electropolished dental implants, largely sacrificing osseointegration due to lower bacterial adhesion. Different passivation methods have been developed to create stable titanium oxide layers. The passivation layers obtained by chemical or electrochemical anodization produce an improved corrosion resistance of the dental implant, total cleaning of the titanium surfaces, and a very significant reduction in the release of titanium ion, but they did not show statistically significant differences in bacterial colonization [23,24,25,26,27].

Currently, diseases caused by bacterial proliferation in the mouth such as mucositis, periodontitis, and peri-implantitis are present in 24% of dental implants, which is a very high proportion. Moreover, the treatment of patients with infected implants is complicated and uncomfortable [28]. Although there are different disinfection techniques with a high biofilm reduction capacity, there are many situations where chemical agents and antibiotics are not effective [28], such as glycine [29], TESPSA [30,31], silver nanoparticles [32], and citric acid [33,34,35]. One of the solutions is implantoplasty, which consists of the mechanical machining of the dental implant to remove the biofilm. This approach has some disadvantages, which are the loss of mechanical properties, the potential damage caused by the particles released into the surrounding tissues, and the lack of the re-osseointegration of the machined surface. In other occasions, implantoplasty cannot be performed, and the infected dental implant must be removed, with a new one placed after cleaning the cavity [29]. This solution is often complicated by the fact that the surrounding teeth must be moved to provide sufficient space for the placement of the new dental implant [28].

In this work, the four most common surfaces of dental implants have been used to study the effect of their topography on cell proliferation and differentiation, as well as initial bacterial adhesion, to find the optimal treatment for bone regeneration. To interpret the cellular and bacterial results, studies of roughness, wettability, and surface energy with polar and dispersive components that will be key to the competition between osteoblastic cells and bacteria have been carried out. There are few studies that use the four most common surfaces in dental implants to study human osteoblasts and bacteria and relate the results to the physicochemical and topographical properties of the surfaces. This study attempts to determine the effect of different topographies on biological and microbiological behavior. The hypothesis of this research is that the rough surface presents a greater capacity for osteoblastic differentiation and a greater adhesion of bacterial plaque on its surface.

## 2. Materials and Methods

In Figure 1 can be observed the scheme of the materials and diiferent metfodologies used for this research.

### 2.1. Sample Preparation

Two-hundred titanium discs (5 mm in diameter and 2 mm in width) with different surface treatments were provided by Galimplant (Sarria, Lugo, Spain). The sample calculation was determined by the number of discs needed from each treatment for biological (n = 25 for each treatment) and microbiological studies (n = 5 for each treatment). The roughness tests, wettability, and SEM observation (n = 3 for each treatment) were non-destructive, and the 3 discs were re-used for microhardness test with 2 discs more for each treatment. The total number was 140 discs: 25 × 4 biological tests + 5 × 4 microbiological tests + 5 × 4 microhardness tests = 140. A total of 35 discs were needed for each treatment, but for safety reasons, 15 supplementary discs for each treatment were made in case any of the tests had to be repeated. The commercially pure titanium was grade 3 (Ti: 99.5%, O: 0.3%, Fe: 0.1%, C: 0.05%, N: 0.05%). The surfaces were:Machined (MACH). The discs were machined without any subsequent surface treatment. (n = 50).Grit-blasted (GBLAST). The roughness was obtained by spraying aluminum oxide (Al_2_O_3_) abrasive particles on the titanium surface at a pressure of 2.5 bars and a gun-sample separation of 70 mm. (n = 50).Acid etching (AE). Acid etching was performed with a mixture of 1:1 concentrated HCl and HNO_3_ acids for 45 s. (n = 50).Grit-blasted and acid etching (GBLAST + AE). The blasting was performed with alumina particles (250–450 µm) at a 2.5 bar pressure and a 100 mm distance. Afterwards, they were washed with distilled water and immersed in a 1:1 mixture of HNO_3_ and concentrated HCl for 45 s. (n = 50).

These four surfaces are the most common in dental implants. The different manufacturers defend more or less rough surfaces, justifying the better adhesion, proliferation, and differentiation of osteoblasts for better and faster osseointegration. In addition, shot blasting improves the fatigue life [36,37,38,39] and attempts to offer a surface that inhibits or at least hinders the formation of bacterial biofilm [40,41,42].

The cleaning of the samples for the surface characterization studies: roughness and machinability was carried out with methyl alcohol for 15 min in ultrasound and then in acetone for 5 min. Drying was done with a hot air flow.

Microhardness tests were performed by Vickers hardness with a high-precision Matzsuzawa microhardness tester (Tokyo, Japan), applying a 1 kg load for 15 s. The indenter was a diamond pyramid. The samples indented were 5 for each treatment and in 5 different places for each sample. Therefore, 4 treatments × 5 samples per treatment × 5 locations in each sample gave a total of 100 indentations.

The indentation diagonals were determined, and the Vickers hardness was calculated according to the equation:$$D=\frac{D_{1}+D_{2}}{2}$$
$$HVN=1.854\;\frac{F}{D^{2}}$$
where *D*_1_ and *D*_2_ are the length of the diagonals of the indentation, *D* is the average of diameters, *F* is the load, 1.854 is the geometrical constant of the pyramidal diamond used, and *HVN* is the Hardness Vickers number.

### 2.2. Characterization of the Titanium Disc Surfaces

The evaluation of surface roughness was performed by means of confocal laser scanning microscopy (CLSM; OLS Olympus Lext 3000, Shinjuku, Japan). First, the equipment was verified with the use of a reference sample (Mitutoyo SR 15, Elgoibar, Spain “Precision Reference Specimen”: Ra = 0.43 μm). A total of 3 measurements in 3 samples of each surface were calculated. The surfaces observed by scanning electron microcopy are shown in Figure 2.

The contact angle analysis was performed on n = 3 samples with ultrapure distilled water (Millipore Milli-Q, Merck Millipore Corporation, Darmstadt, Germany) and formamide (Contact Angle System OCA15plus-Dataphysics, Filderstadt, Germany), and the corresponding data were analyzed with SCA20 (Dataphysics, Filderstadt, Germany). Contact angle measurements were made using the sessile drop method. Drops were generated with a micrometric syringe and were deposited over discs. A total of 3 μL of distilled water and 1 μL of formamide were deposited on each sample at 200 μL/min. Finally, the surface energy was determined by applying the Owens, Wendt, Rabel, and Kaelble (OWRK) equation to the wettability values obtained with distilled water and formamide [43,44,45,46].

### 2.3. Cell Viability and Differentiation

For in vitro studies, osteoblastic cells (SaOS-2; ATCC, Manassas, VA, USA) were used for in vitro studies. They were cultured and McCoy’s modified 5A medium, supplemented with 10% fetal bovine serum (FBS, Gibco, New York, NY, USA), 1% penicillin/streptomycin2mM (Invitrogen, Carlsbad, CA, USA), and 1% sodium pyruvate (Invitrogen, Carlsbad, CA, USA) was used. Cultures were grown at 37 °C in a 5% CO_2_ incubator under humidified conditions with n = 25 for each treatment.

Confluent cells were incubated with TrypLE (Invitrogen, Carlsbad, CA, USA) for 1 min in order to detach them from the flask. Subsequently, 5000 cells were seeded on each disc and incubated at 37 °C. After 3 and 21 days after incubation, the samples were washed with PBS and moved onto a new plate to perform the metabolic activity assay using Alamar Blue (Invitrogen-Thermo Fisher Scientific, Waltham, MA, USA), following the protocol. Briefly, the reagent was prepared and pipetted to cover the samples, and the percentage of Alamar Blue reduction was estimated after 4 h of incubation at 37 °C, using the Alamar Blue solution as a blank.

In order to study osteogenic differentiation, the alkaline phosphatase (ALP) activity was determined by the Sensolyte pNPP alkaline phosphatase colorimetric assay (Anaspec, Fremont, CA, USA). The determination of ALP was measured at a wavelength of 495 nm, and the detection was carried out using a conventional ELx800 microplate reader (Bio-Tek Instruments, Inc., Winooski, VT, USA).

### 2.4. Bacterial Adhesion

Four types of bacteria, *S. gordonii* (CECT 804), *S. oralis* (CECT 907), *A. viscosus* (CECT 488), and *E. faecalis* (CECT 795), were used for the bacterial adhesion test, using tryptic soy broth (TSB) for *S. gordonii*, and brain heart infusion (BHI) for the rest of them as the culture media. We used 5 samples per group, and bacterial strains were tested.

The culture media and material were sterilized by autoclaving at 121 °C for 30 min. The samples were sterilized by washing with ethanol for 5 min, followed by three 5 min washes with H_2_O and an ultraviolet light exposure for 15 min on each side of the samples.

The bacteria inoculums were prepared by suspending the bacteria in 5 mL of the corresponding media and letting them grow overnight at 37 °C. Then, the medium was diluted to 600 nm to an optical density of 0.2. The sterile samples were placed in 24-well plates, and they were covered with 700 µL of the diluted bacterial suspension and incubated at 37 °C for 2 h to analyze the bacterial adhesion. As a positive control, 700 µL of bacterial suspension was added to an empty well plate. After this time, the samples were washed with PBS and moved onto a new 24-well plate for metabolic and live–dead assays.

For the metabolic assay, 3 samples and the positive controls were incubated with 650 µL of 25 µg/mL resazurin sodium salt in PBS (Sigma-Aldrich, St. Louis, MO, USA) at 37 °C until the positive control was saturated. Then, 100 µL was used to read the absorbance at 570 and 600 nm, and this was used to calculate the reduction percentage.

In the case of the Live–Dead assay, the remaining 2 samples were stained with LIVE/DEAD^®^ BackLight™ Bacterial Viability Kit solution (Thermo Fisher Scientific, Waltham, MA, USA). Briefly, the two reagents were diluted in a proportion of 1.5 µL of reagent per mL of PBS, and the samples were covered with 650 µL of the solution and incubated for 15 min at 37 °C. After that, the samples were washed twice with PBS and the images were acquired at three different regions using a confocal laser microscope at 64× (Leica Dmi8, Wetzlar, Germany) using excitation/emission wavelengths of 589/615 nm for dead cells and 495/520 nm for live cells.

### 2.5. Statistical Analysis

The number of samples used was obtained using the experimental sample size method. Statistical analysis was performed using MiniTab 17 software (Minitab Inc., State College, PA, USA). Kruskal–Wallis and Mann–Whitney U non-parametric tests were used. Statistical differences were considered with *p* < 0.001.

## 3. Results

### 3.1. Characterization of the Titanium Disc Surfaces

Figure 2 shows the SEM micrographs of the different roughness’s obtained. The main surface properties (i.e., roughness, contact angle, and surface energy) of the study groups are summarized in Table 1.

It can be seen that the acid-etched samples caused an etching on the surface with a similar roughness as the control. The samples treated with GBLAST have a much higher roughness, being able to appreciate steeper valleys and peaks and some remains of alumina particles that have been used to give roughness. The samples treated with GBLAST and followed by AE show that the roughness of the GBLAST is maintained, but that the sharp angles of the roughness are reduced, while a micro-roughness or engraving of the surface can also be seen in the macro-roughness.

X-ray energy dispersive microanalysis studies show in all the samples of GBLAST and GBLAST + AE that the presence of alumina is lower than 4% on the titanium surfaces. No other type of contamination was observed on any other surface.

The microhardness values of the titanium surfaces were studied, with values for the MACH surfaces of 178 HVN (sd 23), for the AE surfaces of 188 HVN (sd 17), for those treated by GTBLAST abrasive spraying of 265 (sd 23), and for the GBLAST + AE of 268 (sd 23). There are significant differences between the MACH and AE surfaces with respect to the GBLAST and GBLAST + AE. This is due to the co-pressure exerted on the projection of abrasive particles. Between MACH and AE, as well as between GBLAST and GBLAST + AE, no statistically significant differences were observed with a *p* < 0.001.

### 3.2. Cell Viability and Osteogenic Differentiation

Figure 3 shows the cell viability at days 3 and 21 using osteoblastic cells. The viability measure has been obtained by determining the AlamarBlue reduction, being proportional to the cell viability on the different surface types. It can be seen that after both 3 and 21 days of culture, there is a significant difference with *p* < 0.001 on the number of cells between MACH and AE when comparing to the GBLAST and GBLAST + AE samples, being higher in the grit-blasted ones.

Figure 4 shows the results of alkaline phosphatase, with an increase in the enzyme indicating the differentiation of osteoblasts. It can be seen that on day 3, the nmols of pNPP are lower than on day 21. After 21 days, the activity of the enzyme increases in all cases, indicating cell differentiation, but being significantly higher (*p* < 0.001) in those samples treated with grit blasting, GBLAST, and GBLAST + AE.

### 3.3. Bacterial Adhesion

The results of the metabolic activity and live–dead assays for each type of bacteria are represented in Figure 5, Figure 6, Figure 7 and Figure 8.

Looking at the metabolic activity, there are no significant differences among any of the types of titanium surface treatment for all the bacterial types with a *p* value < 0.001. However, there seems to be a tendency for the MACH-treated discs to have a lower bacterial adhesion, even though the differences are not big enough to be statistically significant. When looking at the live–dead images, the results point at a similar number of bacteria in all of the surfaces for all types of bacteria, although in most cases, MACH points at a slight smaller number of bacteria adhered on the surface, which was also seen in the results of the metabolic activity.

## 4. Discussion

The characterization of the differently treated titanium showed similar behavior between the machined and AE samples regarding the roughness, wettability, and surface energy, with no significant differences in any case. In the same way, GBLAST and GBLAST + AE surfaces show very similar properties. It can be seen that the grit blasting of the titanium produces changes in wettability, making them more hydrophobic, but with lower surface energy values [47,48,49,50].

From the roughness results and the wettability and surface energy properties, it can be determined that the machined surfaces and those treated by acid etching have very similar behaviors, with no significant differences in the surface properties in any case. In the same way, GBLAST and GBLAST + AE surfaces show very similar behaviors. It can be seen how the grit blasting treatment produces a change in wettability, producing slightly more hydrophobic surfaces, but with lower surface energy values. We observe from the water energy values significant increases for grit blasting surfaces, which causes an increase in the polar component that will favor osseointegration [50,51,52,53].

Another important difference is in the microhardness values of the surfaces. This fact is due to the compressive residual stresses of the abrasive projection to roughen the surface, which generates compressive values estimated by the authors at −200 MPa. These compressive stresses will be very important for dental implants to achieve a good fatigue resistance since the compressive surface layer will make fatigue crack nucleation very difficult [36,37,38].

In the results, we have been able to verify the presence of alumina in the surfaces treated by grit blasting, which in this case was 4%. This contamination has been estimated by various authors to be up to 8%, and it has been observed that in no case does it affect osseointegration; on the contrary, it causes greater polar behavior which induces greater adhesion, proliferation, and osteoblastic differentiation, generating a higher index of bone in contact with the implant. In the same way, it has been shown that alumina debris also has a certain bactericidal character due to the fact that aluminum oxide has a certain oxidizing effect that is detrimental to bacteria [50,54,55,56,57].

The treatment of the discs influenced both cell proliferation as well as the osteogenic differentiation of SaOs-2 cells. In this case, those with a higher roughness, GBLAST and GBLAST + AE, show improved cell growth and differentiation compared to those with a lower surface roughness. Previous studies have proved the effect of general surface roughness on the behavior of osteogenic cells, improving it with higher roughness values [58,59,60]. The Ra values of GBLAST and GBLAST + AE are both near 2 µm and, precisely, a previous study presented that moderate roughness gradients in the range of 2–3 µm showed an increased osteogenic differentiation in osteoblasts, which can be seen in our study [61,62].

Regarding the initial bacterial adhesion on the different surfaces, no significant differences were found between the different treatments, with a tendency for the MACH samples to have a lower number of adhered bacteria after 2 h of culture, but without being significant. Similar observations were obtained in previous studies using sandblasting and acid etching techniques with resultant comparable Ra values, where they did not observe any significant differences compared to untreated or polished titanium after 2 h of adhesion [63,64,65,66]. However, several studies concluded that increasing the Ra from 34.57 nm to 155 nm or higher produced an increase in the bacterial adhesion after 1 h [27,28,32], but no differences were observed between 155 nm and 223.24 nm or 449.42 nm. This could point to higher bacterial adhesion compared to very smooth titanium samples, but in our case, the MACH titanium already had a significant roughness, therefore not showing a significant differences compared to the rest of the samples regardless of their increased roughness. New treatments allow for a modification of surface properties, such as argon plasma and other spraying techniques, that lead to improvements in biological activity [67,68,69,70].

As we have been able to verify, the hypothesis is fulfilled for the different surfaces studied. We have been able to verify that the rough surfaces obtained by sand-blasting are less hydrophilic surfaces than the control and acid-treated surfaces. However, they have a lower surface energy in both the polar and dispersive components. In principle, the degree of hydrophilicity should increase the degree of osseointegration, but the rough topography favors adhesion, proliferation, and osteoblastic differentiation as well as lower internal energy. This fact also favors the adhesion of the different bacterial strains we have studied and should therefore be taken into account for dental implant designs.

The osteoblastic activity of the different surfaces as well as the bacterial activity could be verified. However, one of the limitations is that we have worked with specific bacterial strains and, in reality, the implants present biofilms, bacterial colonies that are protected by a polysaccharide. Therefore, the results of the study do not reflect reality but they rather show a trend in bacterial activity. The cellular studies reflect better reality, although the influence of the mechanical load that the titanium implants present in the mouth is missing in this study. This fact generally influences a greater osseointegration capacity, but it should be considered as a limitation.

## 5. Conclusions

Surfaces treated by grit blasting show higher roughness values than the control and acid etching. This increase in roughness causes a lower hydrophilicity and a decrease in surface energy. Additionally, once the surface is treated by grit blasting, the subsequent acid etching treatment has little effect on the roughness values and surface energy, with no statistically significant differences being observed. Higher roughness values lead to a decrease in the hydrophilic capacity and lower surface energy. The loss of hydrophilicity could mean a decrease in the osteoblastic activity. However, it is the topography factor that favorably affects osteoblast differentiation, with higher alkaline phosphatase levels. These differences are statistically significant. However, bacterial proliferation in the different strains studied does not show statistically significant differences due to the smaller size of the bacteria with respect to the cells, which makes them insensitive to the topography obtained on the four surfaces studied.

## Figures and Tables

**Figure 1 materials-16-03553-f001:**
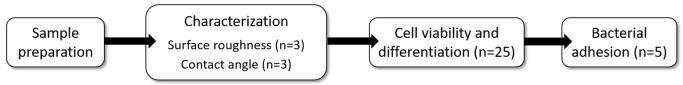
Scheme of the materials and methods used in this research.

**Figure 2 materials-16-03553-f002:**
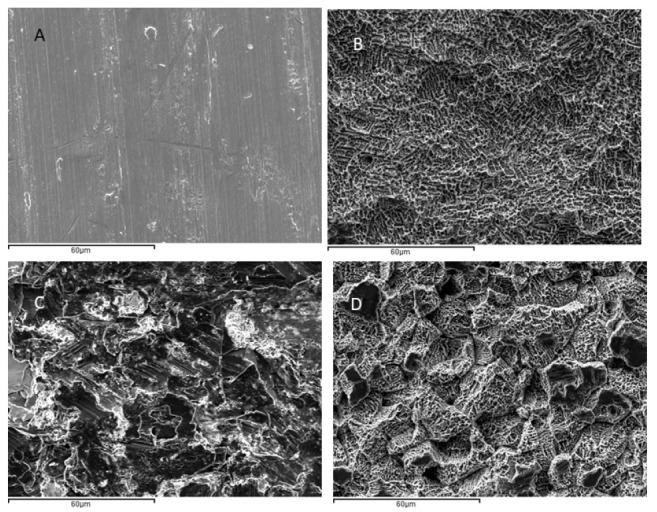
Surfaces observed by SEM of the four treatments studied. (**A**) Machined (MACH), (**B**) acid etching (AE). (**C**) Grit Blasting with alumina (GBLAST). (**D**) Grit Blasting with acid etching (GBLAST + AE).

**Figure 3 materials-16-03553-f003:**
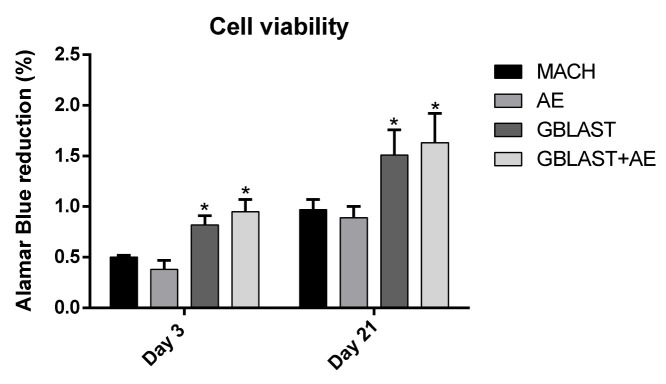
SaOS-2 osteoblastic cells viability for the different topographies studied at 3 and 21 days. Statistical differences at each timepoint are represented with * (*p* < 0.001).

**Figure 4 materials-16-03553-f004:**
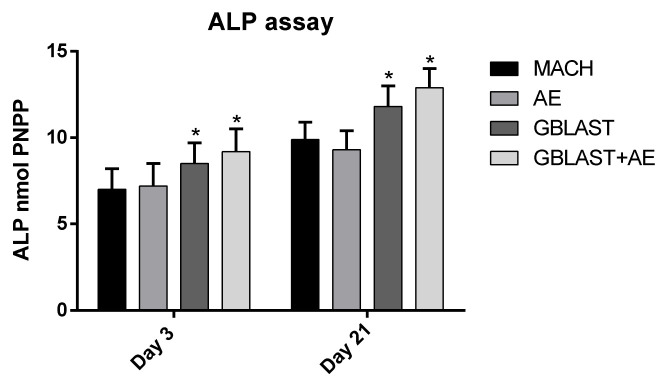
Alkaline phosphatase of SaOS-2 osteoblastic cells for the different topographies studied at 3 and 21 days. Statistical differences at each timepoint are represented with * (*p* < 0.001).

**Figure 5 materials-16-03553-f005:**
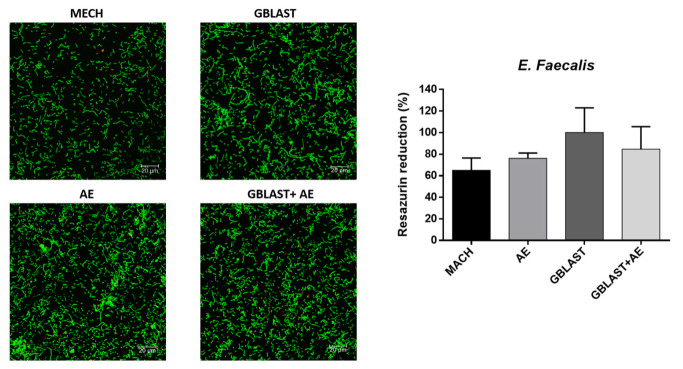
Metabolic activity assay and live–dead images of *E. faecalis*. The green color corresponds to the live cells and red to the dead cells.

**Figure 6 materials-16-03553-f006:**
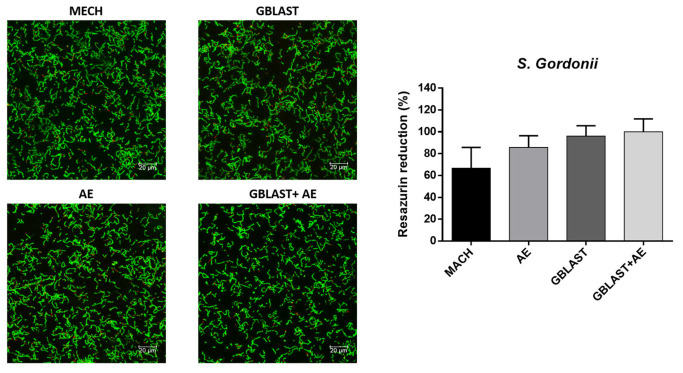
Metabolic activity assay and live–dead images of *S. gordonii*. The green color corresponds to the live cells and red to the dead cells.

**Figure 7 materials-16-03553-f007:**
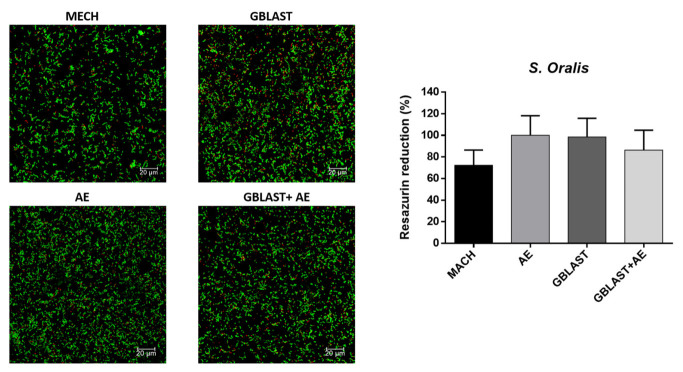
Metabolic activity assay and live–dead images of *S. oralis*. The green color corresponds to the live cells and red to the dead cells.

**Figure 8 materials-16-03553-f008:**
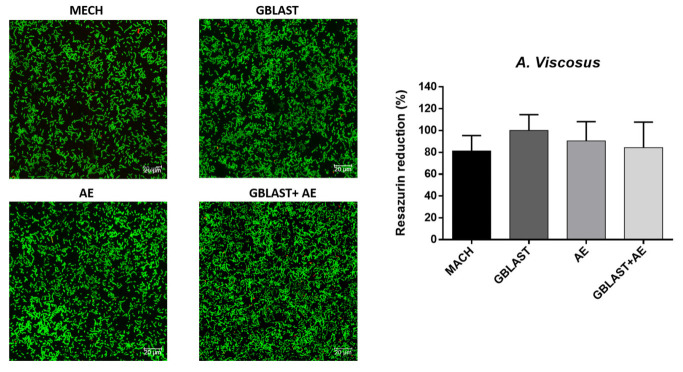
Metabolic activity assay and live–dead images of *A. viscosus*. The green color corresponds to the live cells and red to the dead cells.

**Table 1 materials-16-03553-t001:** Description of surface roughness, contact angle, and surface energy are shown in mean ± standard deviation. Asterisks mean statistical difference significance with *p* < 0.001.

Surface Name	Roughness (Ra) (µm)	Contact Angle (◦)		Surface Energy (mJ/m^2^)
		H_2_O	Formamide	
MACH	0.20 ± 0.06 *	53.4 ± 6.1 *	31.6 ± 4.3 *	49.6 ± 3.3 *
AE	0.35 ± 0.07 *	59.4 ± 2.2 *	36.6 ± 6.2 *	46.5 ± 3.5 *
GBLAST	1.99 ± 0.19 **	89.5 ± 9.9 **	63.2 ± 10.3 **	38.8 ± 4.0 **
GBLAST + AE	2.13 ± 0.15 **	92.3 ± 4.9 **	70.2 ± 12.3 **	39.3 ± 2.7 **

## Data Availability

The authors can provide details of the research requesting by letter and commenting on their needs.

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
