# Peer review of "Relevant Aspects of Titanium Topography for Osteoblastic Adhesion and Inhibition of Bacterial Colonization"

_materials, 2023, doi:10.3390/ma16093553_

Round 1

Reviewer 1 Report

Introduction

- An introduction on the history and develop of different surfaces and methodologies is provided. However, the state of the art about other studies who assessed the biocompatibility of different titanium surfaces is currently missed. I ask you to add a paragraph on that state of the art. For this propose, discuss and cite a very recent published article on the topic:

Alovisi, M.; Carossa, M.; Mandras, N.; Roana, J.; Costalonga, M.; Cavallo, L.; Pira, E.; Putzu, M.G.; Bosio, D.; Roato, I.; Mussano, F.; Scotti, N. Disinfection and Biocompatibility of Titanium Surfaces Treated with Glycine Powder Airflow and Triple Antibiotic Mixture: An In Vitro Study. Materials 202215, 4850. https://doi.org/10.3390/ma15144850

- Add the study hypothesis after the aim of the study

Materials and methods

- This section is currently lacking in fluency. I suggest you to better describe the sequence and the rational. For example, it appears that you first test the biocompatibility and then the bacteria adhesion? why? 

- Did you perform all the test on all the sample (n= 50 per group)? or did you divide the 200 samples between the different tests?  Please clarify

- To make more clear the procedures and the samples division, provide a flow chart of the study at the beginning of the M&M section

- Add a separate paragraph describing the statistical analysis that was performed

Discussion

- Describe if the study hypothesis were accepted or rejected based on the results.

- in your article you are talking about wettability and surface energy. Discuss in the discussion why these aspects are important for cell adhesion and the different methods that were proposed to improve them. For this propose, discuss and cite the following article doi: 10.3390/biom12091219

I believe that your article can be improved following the comments and that it may be accepted after revisions. 

Author Response

REVIEWER 1

Dear Reviewer,

Thanks for taking the time to review our manuscript and suggest to us to improve our work by providing a lot more detail. We have done so, and we are now submitting a manuscript that not only addresses the points the you specifically raised but also many others that we have considered in order to deliver what we think is a much improved version of our work. This version includes more paragraphs, English grammar revisions in all main sections, new references. Thanks a lot. We are looking forward to your comments.

Sincerely,

Francisco-Javier Gil Mur

Introduction

- An introduction on the history and develop of different surfaces and methodologies is provided. However, the state of the art about other studies who assessed the biocompatibility of different titanium surfaces is currently missed. I ask you to add a paragraph on that state of the art. For this propose, discuss and cite a very recent published article on the topic:

Alovisi, M.; Carossa, M.; Mandras, N.; Roana, J.; Costalonga, M.; Cavallo, L.; Pira, E.; Putzu, M.G.; Bosio, D.; Roato, I.; Mussano, F.; Scotti, N. Disinfection and Biocompatibility of Titanium Surfaces Treated with Glycine Powder Airflow and Triple Antibiotic Mixture: An In Vitro Study. Materials 202215, 4850. https://doi.org/10.3390/ma15144850

- A new text has been introduced according to the reviewer and the reference suggested.

Materials and methods

- This section is currently lacking in fluency. I suggest you to better describe the sequence and the rational. For example, it appears that you first test the biocompatibility and then the bacteria adhesion? why? 

Because we wanted to first test possible negative effects of the topography on the cell behaviour, as we thought the impact of the topography on the cells was of bigger importance than the cell adhesion.

- Did you perform all the test on all the sample (n= 50 per group)? or did you divide the 200 samples between the different tests?  Please clarify.

The number of samples for each experiment is stated in the materials and methods. 50 were initially prepared, however, not all of them were used in the experiments. We have introduced the number of samples for each experiment

- To make more clear the procedures and the samples division, provide a flow chart of the study at the beginning of the M&M section

Added

- Add a separate paragraph describing the statistical analysis that was performed

Added

Discussion

- Describe if the study hypothesis were accepted or rejected based on the results.

A new paragraph has been introduced explaining the hypotheses confirmation.

- in your article you are talking about wettability and surface energy. Discuss in the discussion why these aspects are important for cell adhesion and the different methods that were proposed to improve them. For this propose, discuss and cite the following article doi: 10.3390/biom12091219.

The reference and explanation about the suggestion has been introduced in the rext

I believe that your article can be improved following the comments and that it may be accepted after revisions. 

Reviewer 2 Report

Dear Authors ,

The concept of the study is good . However, I have some major concerns .

Abstract :

1.       Consider rephrasing the sentences from line 16-21

2.       What is SaOS-2?

3.       How alkanine phosphatase was determined ?

4.       What was the sample size ?

Introduction :

1.       Authors should consider correcting the English language . Many of the sentences are not making any sense.

2.       “A few years ago, however, it became clear that the surfaces should also take into account bacteria in order to avoid peri-implantitis “ . What does this mean?

3.       “Sometimes, implantoplasty cannot be performed and the infected dental implant must be re moved, and the cavity must be cleaned and bone regenerated to recover the bone resorption produced by the bacteria and place a new implant [20-26]. This sentence is not making any sense . How the authors have used so many times and?? Why many sentences have been merged and making it complicated.

4.       References can be split [20-26] accordingly

5.       The authors have not written any literature review of the previous work.

6.       The authors have not written the rationale  of four type of treatment considered

7.       What is the research gap?

8.       This study attempts to determine the effect of different topographies on biological and microbiological behavior “ of what ? research question is incomplete

9.       Overall the introduction lacks review of literature , rationale for conducting the study, research gap  and finally the aim and objective

Material and method

1.       Any institutional ethical clearance was granted for this study ?

2.       How the sample size has been calculated ?

Result

Many of the sentences are merged without making any sense .

Discussion:

The authors should discuss the observation of their study and comparing with other studies in a sequential manner.  

Limitations and future directions are missing

Conclusion :

The conclusion is more like a summary of the study. It should be written in accordance to what has been inferred from the study making a remark.

Overall the concept of the study is good but it lacks scientific presentation. Considering this it cannot be accepted in the present form.

The authors should take help of native speaker or English editing service . 

Author Response

REVIEWER 2

Dear Reviewer,

Thanks for taking the time to review our manuscript and suggest to us to improve our work by providing a lot more detail. We have done so, and we are now submitting a manuscript that not only addresses the points the you specifically raised but also many others that we have considered in order to deliver what we think is a much improved version of our work. This version includes more paragraphs, English grammar revisions in all main sections, new references. Thanks a lot. We are looking forward to your comments.

Sincerely,

Francisco-Javier Gil Mur

Dear Authors,

The concept of the study is good. However, I have some major concerns.

Abstract:

  1. Consider rephrasing the sentences from line 16-21

Done

  1. What is SaOS-2?

Done

  1. How alkaline phosphatase was determined?

The method has been added in the abstract.

  1. What was the sample size?

The sample size has been introduced in the abstract

Introduction:

  1. Authors should consider correcting the English language. Many of the sentences are not making any sense.

We have improved the English.

  1. “A few years ago, however, it became clear that the surfaces should also take into account bacteria in order to avoid peri-implantitis “. What does this mean?

This sentence has been fixed.

  1. “Sometimes, implantoplasty cannot be performed and the infected dental implant must be re moved, and the cavity must be cleaned and bone regenerated to recover the bone resorption produced by the bacteria and place a new implant [20-26]. This sentence is not making any sense. How the authors have used so many times and?? Why many sentences have been merged and making it complicated.

This has been fixed

  1. References can be split [20-26] accordingly

The references have been separated in different methodologies.  

  1. The authors have not written any literature review of the previous work.

According to the reviewer several paragraphs have been introduced in the introduced with results of other authors

  1. The authors have not written the rationale of four type of treatment considered

The authors have explained the selection of the four types of surfaces.

  1. “This study attempts to determine the effect of different topographies on biological and microbiological behavior “of what? research question is incomplete

Changed it

  1. Overall the introduction lacks review of literature, rationale for conducting the study, research gap and finally the aim and objective

This suggestion has been improved.

Material and method

  1. Any institutional ethical clearance was granted for this study?

There are no ethical issues in this study that needed clearance.

  1. How the sample size has been calculated?

The method has been introduced in Materials and Methods.

Result

Many of the sentences are merged without making any sense.

Fixed

Discussion:

The authors should discuss the observation of their study and comparing with other studies in a sequential manner.  

The authors have introduced new result in the discussion of other authors

Limitations and future directions are missing

This aspect has been introduced in the discussion.

Conclusion:

The conclusion is more like a summary of the study. It should be written in accordance to what has been inferred from the study making a remark.

Conclusions have been re-written.

Overall the concept of the study is good but it lacks scientific presentation. Considering this it cannot be accepted in the present form.

Round 2

Reviewer 1 Report

Dear authors,

Thank you fot addressing my points.

Author Response

Thank you very much for your help and dedication

Reviewer 2 Report

Dear Authors, 

The sample size calculation has not been elaborated.How it was determined? what formula has been used . kindly add this in methodology section

Why in the microhardness test only 20 samples were tested ? "The samples indented were 5 for each treatment" 

How it was determined ? what formula has been used . kindly add this in methodology section 

Minor spell checks are needed. 

Author Response

Thank you again for your suggestions and your patience. 
The authors have explained in detail in Materials and Methods the calculations for the number of samples as suggested by the reviewer. 
In addition, we have added in the same section of Materials and Method.s the equations used for the determination of the Vickers microhardness. English has been revised again
We are at your disposal for any further clarification and again thank you very much.